# Association of preoperative body mass index with postoperative complications and survival for patients with gastric cancer: A systematic review and meta-analysis

Zhenzhen Li ⊙ *, Lili Cui, Jing Sun, Wanlu Liu

Department of Nursing, the Affiliated Taizhou People's Hospital of Nanjing Medical University, Taizhou School of Clinical Medicine, Nanjing Medical University, Taizhou, China

* zzlizhen2023@163.com

**Data Availability Statement:** All relevant data are within the paper and its Supporting Information files.

## Abstract

### Objective

The relationship among body mass index (BMI), postoperative complications, and clinical outcomes in patients undergoing gastrectomy for gastric cancer remains unclear. This study aimed to evaluate this association using a meta-analysis.

### Method

We conducted a systematic search of the PubMed, Embase, and Cochrane Library databases up to February 25, 2024. Patients were classified into underweight ($<18.5$ kg/m$^2$), normal weight ($18.5–25.0$ kg/m$^2$), and overweight ($\geq25.0$ kg/m$^2$) groups based on BMI categories. Meta-analysis was performed using a random-effects model. Additionally, exploratory sensitivity and subgroup analyses were performed.

### Results

Twenty-two studies involving 41,144 patients with gastric cancer were included for quantitative analysis. Preoperative underweight (odds ratio [OR]: 1.26; 95% confidence interval [CI]: 1.03–1.55; $P = 0.024$) and overweight (OR: 1.19; 95%CI: 1.09–1.30; $P < 0.001$) were associated with an increased risk of postoperative complications. Furthermore, preoperative underweight was associated with poorer overall survival (hazard ratio [HR]: 1.40; 95%CI: 1.28–1.53; $P < 0.001$), whereas preoperative overweight was associated with better over-survival (HR: 0.82; 95%CI: 0.73–0.91; $P < 0.001$). Furthermore, preoperative underweight was not associated with disease-free survival (HR: 1.48; 95%CI: 0.97–2.26; $P = 0.069$), whereas preoperative overweight was associated with longer disease-free survival (HR: 0.80; 95%CI: 0.70–0.91; $P = 0.001$). In terms of specific postoperative complications, preoperative underweight was associated with an increased risk of septic shock (OR: 3.40; 95% CI: 1.26–9.17; $P = 0.015$) and a reduced risk of fever (OR: 0.39; 95%CI: 0.18–0.83; $P = 0.014$). Preoperative overweight was associated with an increased risk of wound infections (OR: 1.78; 95%CI: 1.08–2.93; $P = 0.023$), intestinal fistula (OR: 5.23; 95%CI: 1.93–14.21; $P$

**Funding:** The author(s) received no specific funding for this work.

**Competing interests:** The authors have declared that no competing interests exist.

= 0.001), arrhythmia (OR: 6.38; 95%CI: 1.70–24.01; $P$ = 0.006), and pancreatic fistula (OR: 3.37; 95%CI: 1.14–9.96; $P$ = 0.028).

## Conclusion

This study revealed that both preoperative underweight and overweight status were associated with an increased risk of postoperative complications. Moreover, the postoperative survival outcomes were significantly better in overweight compared to that of underweight patients.

## Trial registration

**Registration:** INPLASY202480004.

## Background

Gastric cancer (GC) is the fifth most common malignancy and the third leading cause of cancer-related mortality globally [1]. Anatomically, GC is categorized into two types: cardia and non-cardia cancers, with China contributing to 70% of new cardia cases and 50% of new non-cardia cases worldwide [1,2]. Over recent decades, the global incidence of GC has declined, largely due to successful management of modifiable risk factors such as eradication of *Helicobacter pylori* infection, dietary improvements, regulation of body mass index (BMI), and reduction in smoking and alcohol consumption [3,4]. Currently, surgery is the primary treatment for GC [5,6]. Moreover, factors influencing the prognosis of GC have garnered considerable attention, contributing to a more precise preoperative risk assessment. Four primary risk factors for GC include preoperative carcinoembryonic antigen levels and cancer antigen 19–9 levels, systemic inflammation, and perioperative blood transfusion. These factors have been proven to be significantly correlated with the prognosis of patients with GC [7,8]. In addition to these established risk factors, the health condition of patients with GC is also an important factor affecting their prognosis [9].

Accumulating evidence suggests that nutritional status is associated with the prognosis of various types of cancer [10–12]. Additionally, BMI is a commonly used measure in population-level studies, which provides a general indication of body mass relative to height. Numerous studies have identified a low BMI as an important predictive factor for poor prognosis in patients [13–15]; however, some studies have reported no association between BMI and oncological outcomes [16,17]. Notably, patients with excessive obesity tend to have lower survival rates. However, several studies have revealed that obese patients with cancer have longer survival periods [18–20]. Fat and muscle secrete various hormones and cytokines that may influence the survival of patients with cancer based on body composition [21,22]. These inconsistent results are not solely attributable to variations in patient inclusion and BMI classification. Thus, the relationship between BMI and GC prognosis necessitates further investigation. In this study, we conducted a systematic review and meta-analysis to elucidate the potential association between preoperative BMI and the risk of postoperative complications, and survival outcomes in patients with GC.

## Methods

### Search strategy and selection criteria

The Meta-analysis of Observational Studies in Epidemiology protocol checklist [23] and Preferred Reporting Items for Systematic Reviews and Meta-analyses guidelines [24] were used to conduct and report the meta-analysis. This study was conducted according to the Preferred Reporting of Systematic Reviews and Meta-analyses (PRISMA) guidelines and was registered on the INPLASY platform (No. INPLASY202480004). Studies published in English and Chinese on the association between preoperative BMI, postoperative complications, and survival outcomes in patients with GC met the inclusion criteria without restrictions on publication status. We searched the PubMed, EmBase, and Cochrane Library databases, selecting studies meeting the criteria published up to February 25, 2024, using the following search terms: "gastric cancer," "underweight," "overweight," "obesity," and "body mass index." The reference lists of the retrieved studies were manually reviewed to identify new potentially relevant studies.

The study selection process was conducted independently by two reviewers, and any disagreements between them were resolved by discussion until a consensus was reached. Studies were selected on the basis of their titles, designs, exposures, and outcomes. Studies were included based on the following criteria. (1) Patient demographics: all participants were diagnosed with GC and underwent surgery. (2) Exposure: this included patients who were preoperatively underweight, with a BMI less than 18.5 kg/m$^2$, and those preoperatively overweight, with a BMI of 25.0 kg/m$^2$ or more. (3) Control group: the control group comprised individuals with normal weight, defined as those with a BMI of 18.5–25.0 kg/m$^2$. (4) Outcomes assessed: we focused on postoperative complications, overall survival (OS), and disease-free survival (DFS). (5) Study design: there were no limitations in the type of study design, encompassing both prospective and retrospective cohorts.

### Data collection and quality assessment

The information and data were extracted by one reviewer and verified by the other. The details of the abstracted information included the first author's name, publication year, region, study design, inclusion period, sample size, age, proportion of men, BMI categories, type of gastrectomy, tumor node metastasis stage, reported effect estimates, and outcomes. The quality of the included studies was evaluated using the Newcastle-Ottawa Scale (NOS), with a maximum score of 9 for each individual study. Studies scoring 0–3, 4–6, and 7–9 were categorized as low, moderate, and high quality, respectively [25]. Any discrepancies in in data collection and quality assessment between reviewers were resolved by a third reviewer who consulted the original article.

### Statistical analysis

The association between preoperative BMI and the risk of postoperative and specific complications was assigned as a categorical variable, and odds ratios (OR) with 95% confidence intervals (CI) were calculated based on the crude data before data pooling. The relationship between preoperative BMI and OS or disease-free survival (DFS) was calculated based on the effect estimates in each study, and hazard ratios (HR) with 95% CIs were calculated. Subsequently, a random-effects model was used to pool effect estimates regarding the association between preoperative BMI, postoperative complications, and survival outcomes, considering the underlying variations among the included studies [26,27]. Statistical heterogeneity was assessed using $I^2$ and Cochran's Q statistics, and significant heterogeneity was defined as $I^2$ >50.0% or $P$ <0.10 [28,29]. Sensitivity analyses were performed for postoperative

complications, OS, and DFS to assess the robustness of the pooled conclusions by sequentially removing each study [30]. Furthermore, subgroup analyses were performed for postoperative complications, OS, and DFS according to country, age, proportion of male, study quality, and interaction, and the *P* test with a ratio of effect estimates was used to compare differences between subgroups [31]. Publication bias was assessed using both qualitative and quantitative methods, including funnel plots and the Egger-Begg test results [32,33]. All *P* values reported in this study were two-sided, with a significance level of 0.05. Statistical analyses were performed using the Stata software (version 10.0; StataCorp, College Station, Texas, USA).

## Results

### Search results

An electronic search yielded 1,724 articles, of which 1,246 were included after eliminating duplicates. In total, 1,188 studies were excluded because they reported irrelevant topics. 58 studies were downloaded for full-text evaluation, and 36 were discarded because they reported GC risk (n = 16), other BMI categories (n = 13), or the same population (n = 7). Ultimately, 22 retrospective cohort studies were included in the meta-analysis [13–15,17,19,34–50], and no additional eligible studies were identified after manually reviewing the reference lists of the retrieved studies (Fig 1).

### Study characteristics

Table 1 summarizes the baseline characteristics of the included studies and patients. All studies were retrospective cohorts involving 41,144 patients with GC. Nineteen studies were conducted in Eastern countries (China, Japan, and Korea), whereas the remaining three were conducted in Western countries (USA and Italy). The mean age of the patients included in the studies ranged from 55.4–78.5 years, and the proportion of male in each study varied from 52.0–78.9%. The quality of the studies was assessed using the NOS, with 3, 13, and 6 studies receiving scores of 8, 7, and 6, respectively (S1 Table).

### Postoperative complications

Fourteen studies reported an association between preoperative BMI and the risk of postoperative complications. Preoperative underweight (OR: 1.26; 95%CI: 1.03–1.55; *P* = 0.024) and overweight (OR: 1.19; 95%CI: 1.09–1.30; *P* <0.001) were associated with an elevated risk of postoperative complications (Fig 2). There was significant heterogeneity in preoperative underweight status ($I^2$ = 48.6%; *P* = 0.021), whereas there was not in preoperative overweight status ($I^2$ = 12.1%; *P* = 0.320). Sensitivity analysis revealed that the association between preoperative underweight status and risk of postoperative complications varied, whereas the relationship between preoperative overweight status and postoperative complications remained robust (S1 and S2 Figs).

For specific postoperative complications, preoperative underweight was associated with an increased risk of septic shock (OR: 3.40; 95%CI: 1.26–9.17; *P* = 0.015) and a reduced risk of fever (OR: 0.39; 95%CI: 0.18–0.83; *P* = 0.014). Moreover, preoperative overweight was associated with an increased risk of wound infections (OR: 1.78; 95%CI: 1.08–2.93; *P* = 0.023), intestinal fistula (OR: 5.23; 95%CI: 1.93–14.21; *P* = 0.001), arrhythmia (OR: 6.38; 95%CI: 1.70–24.01; *P* = 0.006), and pancreatic fistula (OR: 3.37; 95%CI: 1.14–9.96; *P* = 0.028). No other significant associations were observed between preoperative BMI and the risk of specific postoperative complications (Table 2).

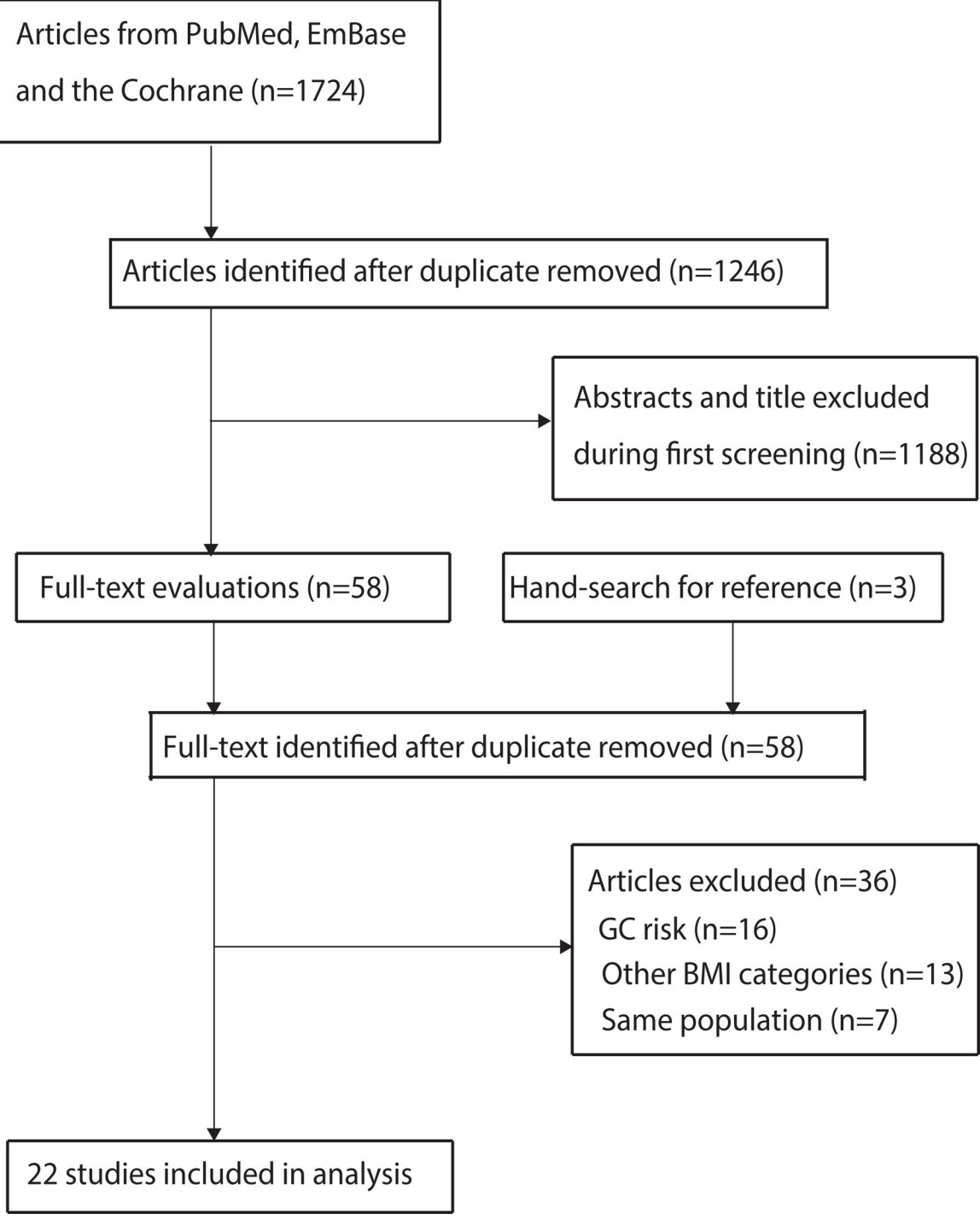

**Fig 1. Literature search and study selection details.**

Subgroup analyses of the association between preoperative BMI and risk of postoperative complications were also performed, as shown in Table 3. We noted that preoperative under-weight status was associated with an elevated risk of postoperative complications when pooled studies were conducted in Eastern countries; the mean age of the patients was <65.0 years,

**Table 1. The baseline characteristics of included studies and involved patients.**

| Study | Region | Study design | Inclusion period | Sample size | Age (years) | Male (%) | BMI categories (kg/m$^2$) | Type of gastrectomy | TNM stage | Study quality |
|---|---|---|---|---|---|---|---|---|---|---|
| Pacelli 2008 [34] | Italy | Retrospective | 2000–2006 | 196 | 65.5 | 61.2 | < 18.5; 18.5–24.9; 25.0–29.9; ≥30.0 | Exploratory laparotomy (27); gastroenteric by-pass (9); total gastrectomy (55); distal subtotal gastrectomy (90); other (15) | I-IV | 6 |
| Nozoe 2012 [35] | Japan | Retrospective | 1998–2010 | 308 | 67.5 | 70.5 | < 18.5; 18.5–24.9; ≥25.0 | Total gastrectomy (100); distal subtotal gastrectomy (208) | I-IV | 7 |
| Yasunaga 2013 [36] | Japan | Retrospective | 2010 | 15582 | 69.1 | 68.6 | < 18.5; 18.5–24.9; 25.0–29.9; ≥30.0 | NA | I-III | 6 |
| Kim 2014 [37] | Korea | Retrospective | 2005–2008 | 304 | 60.0 | 68.1 | < 18.5; 18.5–24.9; 25.0–29.9; ≥30.0 | Total gastrectomy (74); distal subtotal gastrectomy (230) | I-III | 7 |
| Wong 2014 [38] | USA | Retrospective | 1997–2012 | 186 | 67.0 | 52.0 | < 18.5; 18.5–24.9; 25.0–29.9; ≥30.0 | Distal gastrectomy (68); total gastrectomy (64); proximal gastrectomy (2); wedge resection (1) | I-III | 7 |
| Chen 2015 [19] | China | Retrospective | 2000–2010 | 1248 | 58.0 | 72.8 | < 18.5; 18.5–24.9; ≥25.0 | Distal gastrectomy (681); total gastrectomy (329); proximal gastrectomy (238) | I-IV | 7 |
| Ejaz 2015 [39] | USA | Retrospective | 2000–2012 | 775 | 65.9 | 57.5 | < 18.5; 18.5–24.9; 25.0–29.9; ≥30.0 | Total gastrectomy (462); distal subtotal gastrectomy (313) | I-III | 7 |
| Wada 2015 [40] | Japan | Retrospective | 2001–2005 | 427 | 66.0 | 70.3 | < 18.5; 18.5–24.9; ≥25.0 | Total gastrectomy (126); distal subtotal gastrectomy (301) | I-IV | 7 |
| Migita 2016 [41] | Japan | Retrospective | 2003–2011 | 638 | 67.1 | 72.6 | < 18.5; 18.5–24.9; ≥25.0 | NA | I-III | 7 |
| Lee 2016 [42] | Korea | Retrospective | 2000–2008 | 1909 | 58.3 | 67.9 | < 18.5; 18.5–24.9; ≥25.0 | Distal gastrectomy (1354); total gastrectomy (550); proximal gastrectomy (1) | I-IV | 6 |
| Feng 2018 [43] | China | Retrospective | 2008–2015 | 1210 | 59.0 | 78.4 | < 18.5; 18.5–24.9; ≥25.0 | Distal gastrectomy (32); total gastrectomy (114); proximal gastrectomy (7) | I-III | 8 |
| Lee 2018 [13] | Korea | Retrospective | 2000–2016 | 7765 | 58.6 | 66.4 | < 18.5; 18.5–24.9; 25.0–29.9; ≥30.0 | Total gastrectomy (1865); subtotal gastrectomy (5534); functional gastrctomy (366) | I-III | 6 |
| Park 2018 [44] | Korea | Retrospective | 2009–2013 | 2063 | 60.1 | 68.3 | < 18.5; 18.5–24.9; ≥25.0 | Distal gastrectomy (1603); total gastrectomy (334); proximal gastrectomy (105); pylorus-preserving gastrectomy (21) | I-III | 8 |
| Kim 2018 [45] | Korea | Retrospective | 2004–2010 | 510 | 60.9 | 66.5 | < 18.5; 18.5–24.9; ≥25.0 | Total gastrectomy (111); subtotal gastrectomy (399) | I-IV | 7 |
| Wang 2018 [46] | China | Retrospective | 2011–2016 | 827 | 61.4 | 72.1 | < 18.5; 18.5–24.9; ≥25.0 | NA | I-IV | 7 |
| Han 2018 [47] | China | Retrospective | 2007–2010 | 788 | 59.0 | 77.5 | < 18.5; 18.5–24.9; ≥25.0 | NA | I-IV | 6 |
| Zhang 2019 [48] | China | Retrospective | 2015–2017 | 426 | 55.4 | 66.9 | < 18.5; 18.5–24.9; ≥25.0 | Distal gastrectomy (365); total gastrectomy (45); proximal gastrectomy (16) | I-IV | 6 |
| Park 2020 [14] | Korea | Retrospective | 2006–2010 | 1868 | 57.8 | 67.1 | < 18.5; 18.5–24.9; 25.0–29.9; ≥30.0 | Total gastrectomy (708); subtotal gastrectomy (1160) | II-III | 7 |
| Miyasaka 2020 [49] | Japan | Retrospective | 2004–2018 | 440 | 66.3 | 69.3 | < 18.5; 18.5–24.9; ≥25.0 | Total gastrectomy (440) | I-III | 7 |
| Zhao 2021 [17] | China | Retrospective | 2003–2011 | 871 | NA | 74.7 | < 18.5; 18.5–24.9; ≥25.0 | Distal gastrectomy (469); total gastrectomy (258); proximal gastrectomy (144) | I-III | 7 |

*(Continued)*

**Table 1.** (Continued）

| Study | Region | Study design | Inclusion period | Sample size | Age (years) | Male (%) | BMI categories (kg/m$^2$) | Type of gastrectomy | TNM stage | Study quality |
|---|---|---|---|---|---|---|---|---|---|---|
| Ma 2021 [15] | China | Retrospective | 2013–2018 | 2526 | NA | 78.9 | < 18.5; 18.5–24.9; 25.0–29.9; ≥30.0 | Radical gastrectomy (2526) | I-III | 8 |
| Jeong 2023 [50] | Korea | Retrospective | 2007–2015 | 277 | 78.5 | 66.7 | < 18.5; 18.5–24.9; ≥25.0 | Distal gastrectomy (173); total gastrectomy (64) | II-III | 7 |

and the proportion of male patients was <70.0%. Moreover, preoperative overweight was associated with an increased risk of postoperative complications in studies conducted in Eastern countries, particularly when the mean age of the patients was <65 years, irrespective of the proportion of male and study quality. Finally, it was observed that the association between preoperative underweight and the risk of postoperative complications could be influenced by the mean age of patients (OR: 0.67; 95%CI: 0.47–0.97; $P$ = 0.032).

Furthermore, we assessed publication bias regarding the association between preoperative BMI and risk of postoperative complications, as shown in S3 and S4 Figs. We noted no significant association between preoperative underweight ($P$ value for Egger: 0.537; $P$ value for Begg: 0.827) and overweight ($P$ value for Egger: 0.323; $P$ value for Begg: 0.381) and the risk of postoperative complications.

## OS

Thirteen studies reported an association between preoperative BMI and OS. Preoperative underweight was associated with decreased OS (HR: 1.40; 95%CI: 1.28–1.53; $P$ <0.001), whereas overweight was associated with increased OS (HR: 0.82; 95%CI: 0.73–0.91; $P$ <0.001) (Fig 3). There was no significant heterogeneity in preoperative underweight status ($I^2$ = 0.0%; $P$ = 0.820), but there was significant heterogeneity in preoperative overweight status ($I^2$ = 42.9%; $P$ = 0.050). The pooled conclusions regarding the association between preoperative underweight or overweight status and OS remained stable (S5 and S6 Figs).

Subgroup analyses of the association between preoperative BMI and OS are shown in Table 3. We noted that a preoperative underweight status was associated with poor OS in most subgroups, whereas no significant association was observed in pooled studies conducted in Western countries. Furthermore, preoperative overweight was associated with enhanced OS in most subgroups. However, this association was not observed in pooled studies conducted in Western countries involving patients with a mean age of ≥65.0 years and where the proportion of male patients was ≥70.0%.

Furthermore, we assessed the publication bias in the association between preoperative BMI and OS (S7 and S8 Figs). There was no significant publication bias in the association between preoperative underweight ($P$ value for Egger: 0.800; $P$ value for Begg: 0.951) and overweight ($P$ value for Egger: 0.759; $P$ value for Begg: 0.855) and OS.

## DFS

Six studies reported an association between the preoperative BMI and DFS. Preoperative underweight was not associated with DFS (HR: 1.48; 95%CI: 0.97–2.26; $P$ = 0.069), whereas preoperative overweight was associated with increased DFS (HR: 0.80; 95%CI: 0.70–0.91; $P$ = 0.001) (Fig 4). There was significant heterogeneity in preoperative underweight status ($I^2$ = 73.9%; $P$ = 0.002), whereas there was no evidence of heterogeneity in preoperative overweight status ($I^2$ = 0.0%; $P$ = 0.907). Sensitivity analysis revealed that preoperative underweight status

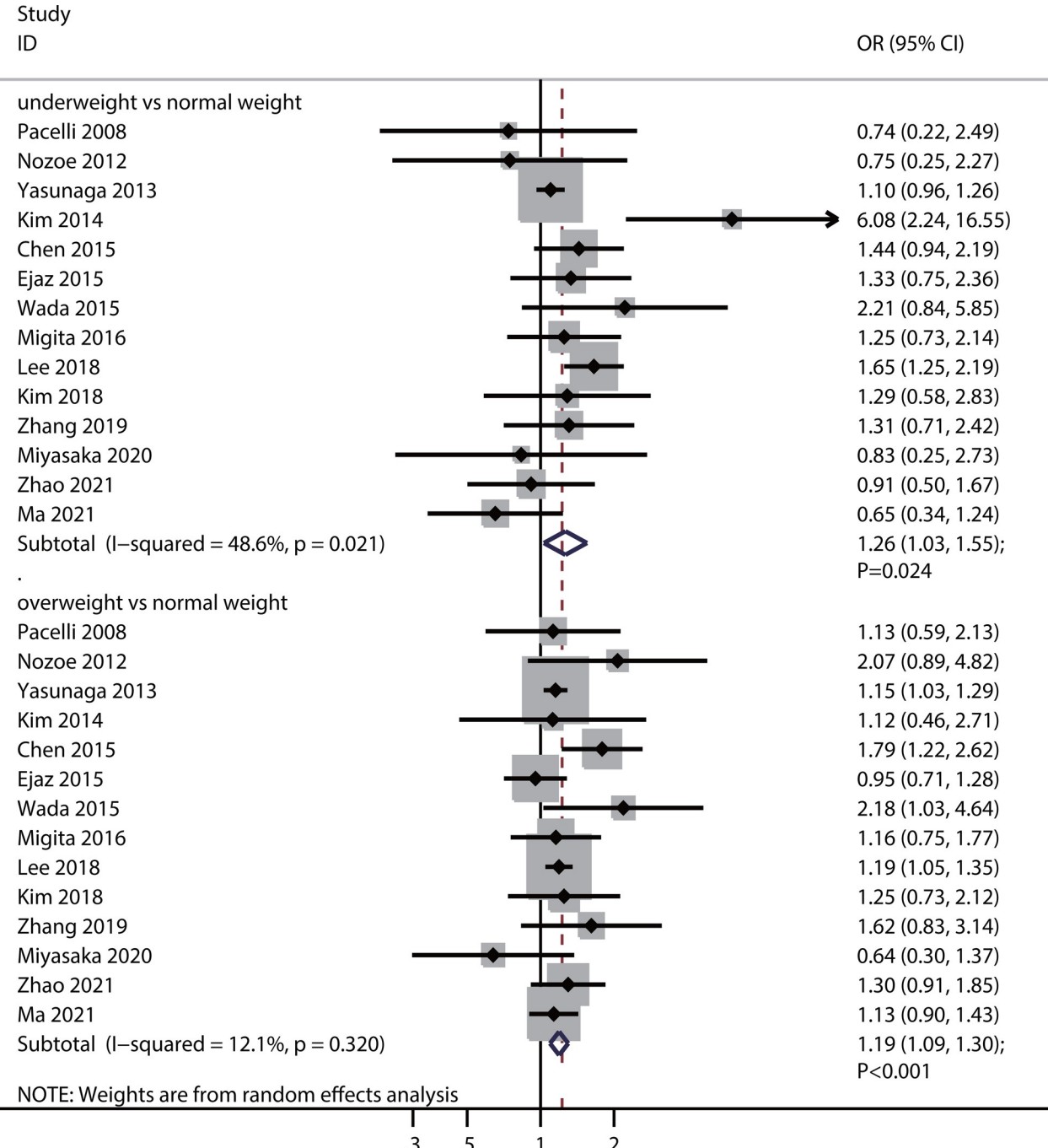

**Fig 2. Association of preoperative BMI with the risk of postoperative complications.**

was associated with poor DFS after excluding the study by Park et al. [44]. In contrast, preoperative overweight status did not affect DFS after excluding the study by Lee et al. [13] (S9 and S10 Figs).

Subgroup analyses on the association between preoperative BMI and DFS are shown in Table 3. Preoperative underweight was correlated with diminished DFS when the proportion of male was ≥70.0%, and the studies assessed were of moderate quality. Furthermore, an

**Table 2. The summary results for specific postoperative complications.**

| Outcomes | BMI categories | Number of studies | OR and 95%CI | *P* value | $I^2$ (%) | *P* value for heterogeneity |
|---|---|---|---|---|---|---|
| Pneumonia | Underweight | 5 | 1.00 (0.41–2.44) | 0.998 | 67.2 | 0.016 |
| | Overweight | 5 | 1.25 (0.91–1.72) | 0.166 | 0.0 | 0.968 |
| Abdominal abscess | Underweight | 3 | 1.38 (0.60–3.20) | 0.451 | 0.0 | 0.611 |
| | Overweight | 3 | 1.02 (0.55–1.90) | 0.951 | 0.0 | 0.873 |
| Septic shock | Underweight | **2** | **3.40 (1.26–9.17)** | **0.015** | **0.0** | **0.388** |
| | Overweight | 2 | 1.24 (0.39–3.93) | 0.718 | 0.0 | 0.344 |
| Anastomotic leak | Underweight | 6 | 1.18 (0.55–2.51) | 0.672 | 0.0 | 0.657 |
| | Overweight | 6 | 1.41 (0.73–2.72) | 0.312 | 18.0 | 0.297 |
| Wound dehiscence | Underweight | 1 | 0.44 (0.02–8.21) | 0.581 | - | - |
| | Overweight | 2 | 2.16 (0.48–9.79) | 0.317 | 0.0 | 0.868 |
| Gastrointestinal perforation, obstruction, and ischemia | Underweight | 1 | 1.61 (0.06–41.17) | 0.774 | - | - |
| | Overweight | 1 | 0.90 (0.06–14.67) | 0.943 | - | - |
| Hemoperitineum | Underweight | 3 | 2.23 (0.71–7.03) | 0.172 | 0.0 | 0.375 |
| | Overweight | 3 | 1.92 (0.51–7.18) | 0.331 | 38.7 | 0.196 |
| Wound infections | Underweight | 3 | 0.77 (0.35–1.69) | 0.521 | 0.0 | 0.942 |
| | Overweight | **3** | **1.78 (1.08–2.93)** | **0.023** | **0.0** | **0.503** |
| Urinary tract infections | Underweight | 3 | 2.35 (0.56–9.92) | 0.246 | 0.0 | 0.394 |
| | Overweight | 3 | 0.81 (0.22–2.93) | 0.749 | 0.0 | 0.890 |
| Nausea | Underweight | 1 | 1.67 (0.66–4.20) | 0.278 | - | - |
| | Overweight | 1 | 1.94 (0.85–4.46) | 0.117 | - | - |
| Gastroparesis | Underweight | 2 | 0.74 (0.26–2.11) | 0.576 | 0.0 | 0.643 |
| | Overweight | 2 | 1.11 (0.59–2.11) | 0.740 | 0.0 | 0.837 |
| Mechanical obstruction | Underweight | 2 | 2.71 (0.23–32.34) | 0.431 | 68.2 | 0.076 |
| | Overweight | 2 | 1.02 (0.41–2.50) | 0.969 | 0.0 | 0.756 |
| Ileus | Underweight | 2 | 1.29 (0.31–5.36) | 0.729 | 30.8 | 0.229 |
| | Overweight | 2 | 1.44 (0.52–3.99) | 0.480 | 0.0 | 0.544 |
| Intestinal fistula | Underweight | 2 | 2.46 (0.30–20.02) | 0.399 | 37.9 | 0.205 |
| | Overweight | **2** | **5.23 (1.93–14.21)** | **0.001** | **0.0** | **0.570** |
| Gastrointestinal hemorrhage | Underweight | 1 | 5.78 (0.36–92.84) | 0.216 | - | - |
| | Overweight | 1 | 10.08 (0.91–111.73) | 0.060 | - | - |
| Diarrhea | Underweight | 1 | 1.14 (0.05–23.94) | 0.931 | - | - |
| | Overweight | 1 | 0.99 (0.05–20.78) | 0.997 | - | - |
| Arrhythmia | Underweight | 1 | 1.44 (0.16–12.96) | 0.745 | - | - |
| | Overweight | **1** | **6.38 (1.70–24.01)** | **0.006** | - | - |
| Pleural effusion | Underweight | 3 | 1.28 (0.16–10.33) | 0.815 | 79.0 | 0.009 |
| | Overweight | 3 | 0.82 (0.35–1.90) | 0.637 | 0.0 | 0.571 |
| ARDS | Underweight | 1 | 3.87 (0.64–23.33) | 0.140 | - | - |
| | Overweight | 1 | 0.71 (0.04–13.78) | 0.820 | - | - |
| Heart failure | Underweight | 1 | 1.14 (0.05–23.94) | 0.931 | - | - |
| | Overweight | 1 | 2.50 (0.23–27.75) | 0.455 | - | - |
| Renal failure | Underweight | 1 | 3.87 (0.64–23.33) | 0.140 | - | - |
| | Overweight | 1 | 1.67 (0.17–16.11) | 0.659 | - | - |
| Liver failure | Underweight | 1 | 5.78 (0.36–92.84) | 0.216 | - | - |
| | Overweight | 1 | 1.66 (0.07–40.85) | 0.757 | - | - |
| CNS complications | Underweight | 1 | 1.92 (0.20–18.59) | 0.573 | - | - |
| | Overweight | 1 | 1.67 (0.17–16.11) | 0.659 | - | - |

(*Continued*)

**Table 2.** (Continued)

| Outcomes | BMI categories | Number of studies | OR and 95%CI | P value | $I^2$ (%) | P value for heterogeneity |
|---|---|---|---|---|---|---|
| Pancreatic fistula | Underweight | 2 | 1.08 (0.13–9.26) | 0.945 | 0.0 | 0.463 |
| | Overweight | **2** | **3.37 (1.14–9.96)** | **0.028** | **0.0** | **0.727** |
| Fever | Underweight | **1** | **0.39 (0.18–0.83)** | **0.014** | - | - |
| | Overweight | 1 | 1.17 (0.68–2.01) | 0.571 | - | - |
| Chyle leakage | Underweight | 1 | 1.61 (0.31–8.43) | 0.572 | - | - |
| | Overweight | 1 | 0.36 (0.02–6.53) | 0.488 | - | - |
| Incisional hernia | Underweight | 1 | 0.28 (0.04–2.07) | 0.211 | - | - |
| | Overweight | 1 | 1.37 (0.70–2.69) | 0.361 | - | - |
| Hemorrhage | Underweight | 1 | 0.87 (0.36–2.11) | 0.754 | - | - |
| | Overweight | 1 | 0.94 (0.54–1.64) | 0.821 | - | - |
| Thrombus | Underweight | 1 | 0.78 (0.10–6.18) | 0.818 | - | - |
| | Overweight | 1 | 0.84 (0.23–3.05) | 0.793 | - | - |

association was found between preoperative overweight and extended DFS in scenarios where studies were conducted in Eastern countries, the average age of patients was <65.0 years, the proportion of male was <70.0%, and the quality of the studies was moderate. Notably, the relationship between preoperative underweight and DFS appeared to be influenced by the proportion of male (OR: 3.50; 95%CI: 1.72–7.13; $P$ = 0.001).

Furthermore, we assessed the publication bias regarding the association between preoperative BMI and DFS, as shown in S11 and S12 Figs. There was no significant publication bias in the association between preoperative underweight ($P$ value for Egger: 0.869; $P$ value for Begg: 0.707) and overweight ($P$ value for Egger: 0.864; $P$ value for Begg: 1.000) with DFS.

## Discussion

This comprehensive quantitative study included 41,144 patients with GC from 22 retrospective cohort studies and analyzed a diverse range of characteristics. We found that preoperative underweight status was associated with an increased risk of postoperative complications and poor OS, but it did not affect DFS. Regarding specific postoperative complications, preoperative underweight status was linked to a higher risk of septic shock and a lower risk of fever. Additionally, preoperative overweight status was associated with an increased risk of postoperative complications and longer OS and DFS. Regarding specific postoperative complications, preoperative overweight status was associated with a high risk of wound infections, intestinal fistulas, arrhythmias, and pancreatic fistulas. Finally, the association among preoperative BMI, postoperative complications, and survival outcomes may have been influenced by factors such as mean age and proportion of male patients.

Several systematic reviews and meta-analyses have addressed the association between preoperative underweight or overweight status and GC prognosis [51,52]. Zhao et al. identified 12 studies and found that underweight was associated with an increased risk of postoperative complications and poor short- and long-term survival outcomes [51]. Another meta-analysis of 36 studies, that applied a BMI cutoff of 25 kg/m$^2$, conducted by the same team, suggested that a high BMI was associated with an increased risk of postoperative complications, particularly infectious complications. However, it did not affect the postoperative mortality or long-term survival of patients with GC [52]. Nonetheless, this study has the following limitations: (1) the association between preoperative underweight status and postoperative outcomes in GC has not been thoroughly explored; (2) the reference group for the prognosis of overweight

**Table 3. Subgroup analyses for postoperative complications, overall survival, and disease-free survival.**

| Outcomes | Factors | Subgroups | BMI categories | Number of studies | OR and 95%CI | P value | I² (%) | P value for heterogeneity | Ratio between subgroups for underweight | Ratio between subgroups for overweight |
|---|---|---|---|---|---|---|---|---|---|---|
| Postoperative complications | Country | Eastern | Underweight | 12 | 1.28 (1.02–1.60) | 0.033 | 55.2 | 0.011 | 1.07 (0.61–1.88); P = 0.823 | 1.23 (0.93–1.64); P = 0.143 |
| | | | Overweight | 12 | 1.21 (1.11–1.33) | < 0.001 | 12.8 | 0.320 | | |
| | | Western | Underweight | 2 | 1.20 (0.71–2.01) | 0.497 | 0.0 | 0.389 | | |
| | | | Overweight | 2 | 0.98 (0.75–1.28) | 0.892 | 0.0 | 0.643 | | |
| | Age (years) | ≥ 65.0 | Underweight | 7 | 1.12 (0.99–1.26) | 0.081 | 0.0 | 0.720 | 0.67 (0.47–0.97); P = 0.032 | 0.88 (0.69–1.12); P = 0.283 |
| | | | Overweight | 7 | 1.13 (0.95–1.36) | 0.164 | 28.5 | 0.211 | | |
| | | < 65.0 | Underweight | 5 | 1.66 (1.19–2.34) | 0.003 | 49.1 | 0.097 | | |
| | | | Overweight | 5 | 1.29 (1.10–1.52) | 0.002 | 12.4 | 0.335 | | |
| | Male (%) | ≥ 70.0 | Underweight | 6 | 1.12 (0.82–1.52) | 0.493 | 29.7 | 0.212 | 0.81 (0.53–1.23); P = 0.320 | 1.19 (0.96–1.48); P = 0.114 |
| | | | Overweight | 6 | 1.37 (1.12–1.68) | 0.002 | 31.4 | 0.200 | | |
| | | < 70.0 | Underweight | 8 | 1.39 (1.03–1.85) | 0.029 | 61.1 | 0.012 | | |
| | | | Overweight | 8 | 1.15 (1.06–1.24) | < 0.001 | 0.0 | 0.638 | | |
| | Study quality | High | Underweight | 10 | 1.27 (0.93–1.74) | 0.138 | 49.5 | 0.037 | 1.00 (0.65–1.53); P = 1.000 | 1.05 (0.87–1.27); P = 0.608 |
| | | | Overweight | 10 | 1.23 (1.04–1.47) | 0.019 | 34.0 | 0.136 | | |
| | | Moderate | Underweight | 4 | 1.27 (0.95–1.70) | 0.103 | 59.1 | 0.062 | | |
| | | | Overweight | 4 | 1.17 (1.08–1.27) | < 0.001 | 0.0 | 0.786 | | |
| Overall survival | Country | Eastern | Underweight | 11 | 1.40 (1.27–1.53) | < 0.001 | 0.0 | 0.702 | 0.97 (0.61–1.54); P = 0.883 | 0.99 (0.60–1.63); P = 0.962 |
| | | | Overweight | 11 | 0.81 (0.72–0.90) | < 0.001 | 41.0 | 0.075 | | |
| | | Western | Underweight | 2 | 1.45 (0.92–2.29) | 0.112 | 0.0 | 0.604 | | |
| | | | Overweight | 2 | 0.82 (0.50–1.35) | 0.435 | 64.8 | 0.092 | | |
| | Age (years) | ≥ 65.0 | Underweight | 4 | 1.57 (1.16–2.12) | 0.003 | 0.0 | 0.866 | 1.14 (0.83–1.57); P = 0.428 | 1.15 (0.78–1.69); P = 0.488 |
| | | | Overweight | 4 | 0.86 (0.59–1.25) | 0.420 | 68.9 | 0.022 | | |
| | | < 65.0 | Underweight | 7 | 1.38 (1.24–1.53) | < 0.001 | 0.0 | 0.495 | | |
| | | | Overweight | 7 | 0.75 (0.68–0.82) | < 0.001 | 0.0 | 0.663 | | |
| | Male (%) | ≥ 70.0 | Underweight | 5 | 1.44 (1.28–1.62) | < 0.001 | 0.0 | 0.767 | 1.07 (0.89–1.28); P = 0.490 | 1.19 (0.98–1.44); P = 0.085 |
| | | | Overweight | 5 | 0.89 (0.78–1.01) | 0.079 | 26.6 | 0.244 | | |
| | | < 70.0 | Underweight | 8 | 1.35 (1.17–1.55) | < 0.001 | 0.0 | 0.633 | | |
| | | | Overweight | 8 | 0.75 (0.65–0.87) | < 0.001 | 36.4 | 0.138 | | |
| | Study quality | High | Underweight | 10 | 1.37 (1.20–1.56) | < 0.001 | 0.0 | 0.622 | 0.96 (0.80–1.15); P = 0.644 | 1.09 (0.91–1.31); P = 0.344 |
| | | | Overweight | 10 | 0.84 (0.73–0.96) | 0.011 | 48.2 | 0.043 | | |
| | | Moderate | Underweight | 3 | 1.43 (1.26–1.62) | < 0.001 | 0.0 | 0.919 | | |
| | | | Overweight | 3 | 0.77 (0.68–0.86) | < 0.001 | 0.0 | 0.451 | | |

(*Continued*)

**Table 3.** (Continued)

| Outcomes | Factors | Subgroups | BMI categories | Number of studies | OR and 95%CI | P value | I² (%) | P value for heterogeneity | Ratio between subgroups for underweight | Ratio between subgroups for overweight |
|---|---|---|---|---|---|---|---|---|---|---|
| Disease free survival | Country | Eastern | Underweight | 5 | 1.47 (0.93–2.32) | 0.098 | 79.2 | 0.001 | 0.92 (0.22–3.86); P = 0.914 | 1.17 (0.67–2.06); P = 0.575 |
| | | | Overweight | 5 | 0.81 (0.71–0.92) | 0.002 | 0.0 | 0.871 | | |
| | | Western | Underweight | 1 | 1.59 (0.41–6.14) | 0.501 | - | - | | |
| | | | Overweight | 1 | 0.69 (0.40–1.19) | 0.179 | - | - | | |
| | Age (years) | ≥ 65.0 | Underweight | 3 | 1.98 (0.65–6.00) | 0.227 | 70.1 | 0.035 | 1.57 (0.49–5.08); P = 0.450 | 0.91 (0.62–1.35); P = 0.652 |
| | | | Overweight | 3 | 0.74 (0.51–1.06) | 0.104 | 0.0 | 0.932 | | |
| | | < 65.0 | Underweight | 3 | 1.26 (0.87–1.84) | 0.217 | 70.4 | 0.034 | | |
| | | | Overweight | 3 | 0.81 (0.70–0.93) | 0.003 | 0.0 | 0.551 | | |
| | Male (%) | ≥ 70.0 | Underweight | 1 | 4.44 (2.34–8.43) | < 0.001 | - | - | 3.50 (1.72–7.13); P = 0.001 | 0.90 (0.31–2.60); P = 0.846 |
| | | | Overweight | 1 | 0.72 (0.25–2.06) | 0.541 | - | - | | |
| | | < 70.0 | Underweight | 5 | 1.27 (0.93–1.73) | 0.127 | 47.3 | 0.108 | | |
| | | | Overweight | 5 | 0.80 (0.70–0.91) | 0.001 | 0.0 | 0.825 | | |
| | Study quality | High | Underweight | 5 | 1.44 (0.73–2.84) | 0.297 | 79.1 | 0.001 | 0.97 (0.47–2.01); P = 0.941 | 1.08 (0.83–1.40); P = 0.585 |
| | | | Overweight | 5 | 0.84 (0.68–1.03) | 0.095 | 0.0 | 0.861 | | |
| | | Moderate | Underweight | 1 | 1.48 (1.14–1.92) | 0.003 | - | - | | |
| | | | Overweight | 1 | 0.78 (0.66–0.92) | 0.003 | - | - | | |

patients with GC preoperatively had a BMI <25 kg/m$^2$, which may have included underweight patients, potentially introducing bias into the association between preoperative overweight status and postoperative outcomes; and (3) new studies meeting the inclusion criteria have been published in recent years, necessitating an update to the meta-analysis outcomes. Thus, the current study aimed to assess the potential association among preoperative BMI, risk of postoperative complications, and survival outcomes.

The results indicated that preoperative underweight status was associated with an increased risk of postoperative complications. Tumor growth may result in significant nutrient depletion and malnutrition-related symptoms, primarily due to systemic inflammatory response induced by the tumor growth, leading to insulin resistance and accelerated catabolism of proteins and adipose tissues. Moreover, studies have demonstrated that preoperative serum albumin levels in underweight patients are lower than those in normal-weight patients, which may contribute to the higher incidence postoperative complications [53,54]. Moreover, preoperative underweight status was associated with poor OS, which may be explained by the fact that underweight patients tend to have later tumor staging and more aggressive tumor invasion than other patients [55]. A preoperative underweight status can result from inadequate nutritional intake, which the aggressive nature of the tumor may exacerbate. However, low body weight may not be an independent prognostic factor in patients with GC [41]. Furthermore, tumor progression can lead to preoperative weight loss and tumor-related malnutrition, increasing the risk of postoperative complications and nutritional deficiencies, which can negatively affect the survival outcomes of patients with GC. Poor nutritional status and increased postoperative complications may delay the initiation of adjuvant chemotherapy and increase the potential toxicity of the chemotherapy drugs [56,57]. These unfavorable conditions are important factors that lead to postoperative recurrence and poor prognosis these patients [58].

Furthermore, this study found that preoperative overweight status was associated with an increased risk of postoperative complications. Being overweight can significantly complicate

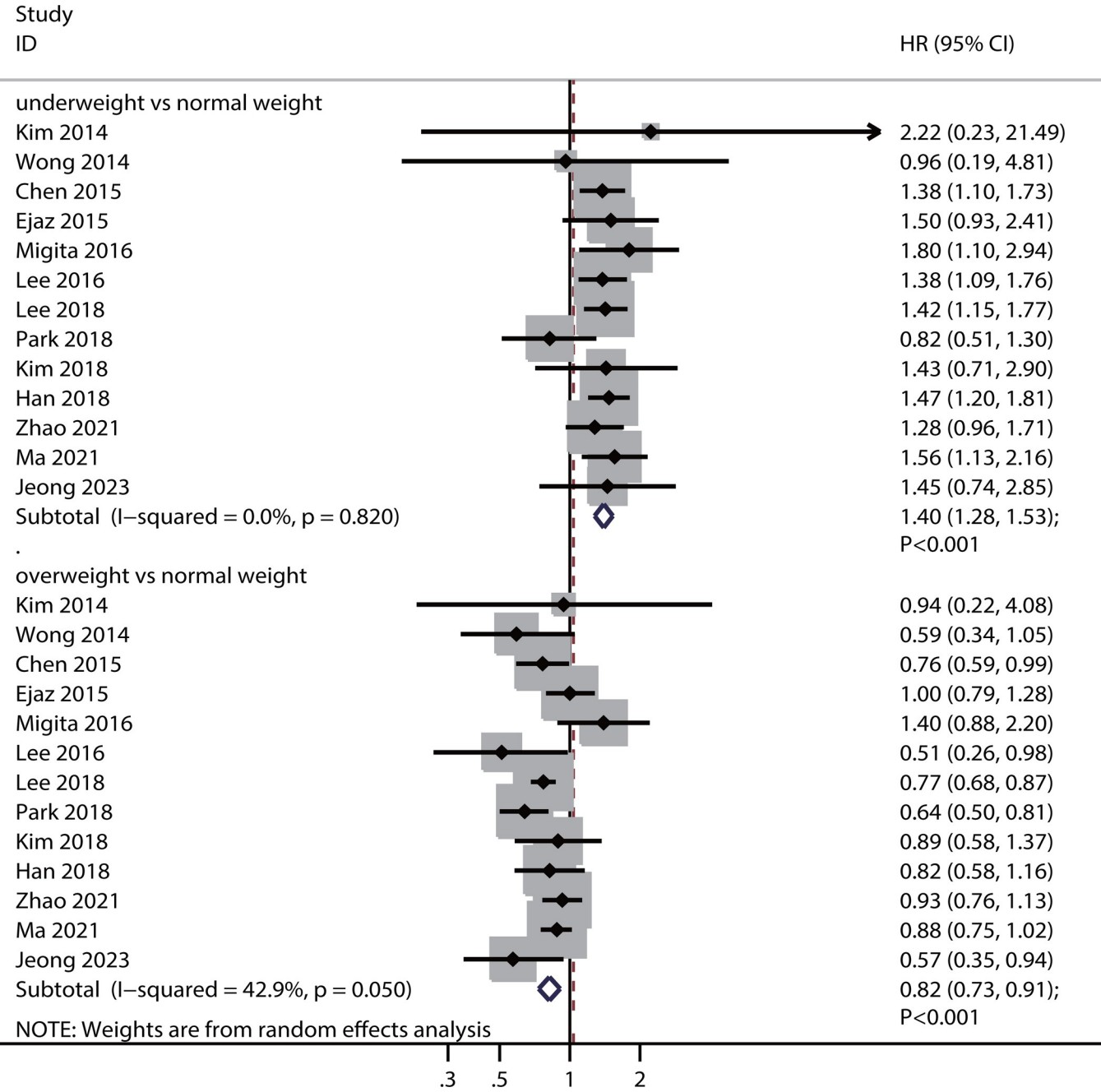

**Fig 3. Association of preoperative BMI with OS.**

GC surgery [59]. Excess intra-abdominal fat may interfere with intraoperative lymph node dissection and hinder the retrieval of lymph nodes from resected specimens. Additionally, it may cover major blood vessels, increasing the technical complexity of the surgery and elevating the risk of intraoperative bleeding. Notably, substantial intraoperative bleeding can further obscure the surgical field, making lymph node dissection challenging. These unfavorable conditions inevitably prolong surgical and anesthesia times, leading to an increased risk of postoperative complications. Excess subcutaneous fat tissue at the incision site, high intra-abdominal

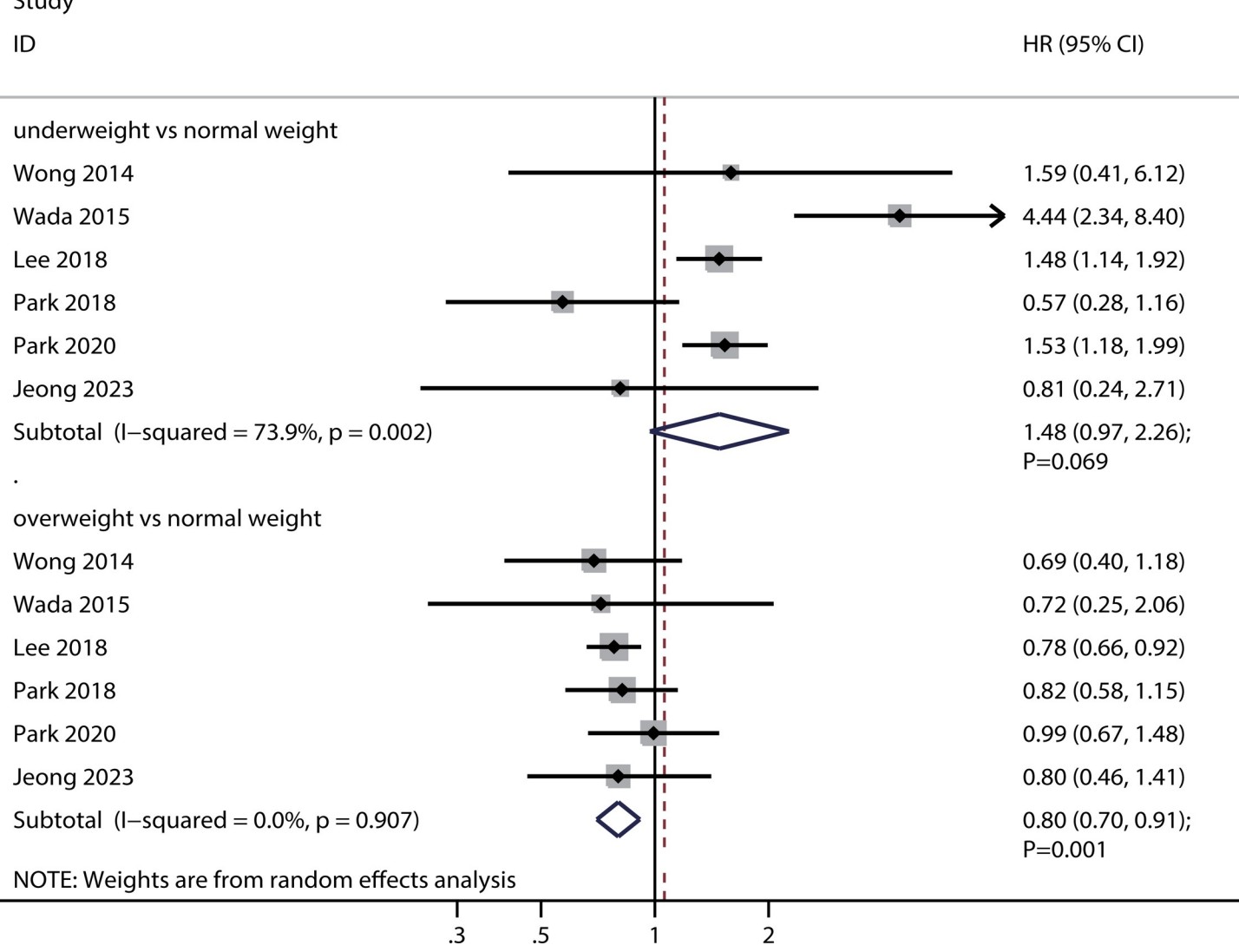

Study

ID                                                                                                                    HR (95% CI)

**Fig 4. Association of preoperative BMI with DFS.**

pressure, and high surface tension may be the reasons for incision-related issues [60]. Additionally, a preoperative overweight status was associated with longer OS and DFS in patients with GC. A preoperative overweight status may be associated with improved OS and DFS, a phenomenon known as the "obesity paradox." The underlying mechanism could involve overweight individuals having greater nutritional reserves and metabolic support, which aid in resisting physiological stress during treatment, as well as anti-inflammatory factors potentially produced by fat tissue, influencing the tumor microenvironment, and indirectly affecting tumor progression [61]. Nonetheless, overweight and obesity remain recognized as risk factors for GC, suggesting that they play distinct roles in the early stages of disease initiation and development [62].

Regarding specific postoperative complications, we found that a preoperative underweight status was associated with an increased risk of septic shock and a reduced risk of fever. Importantly, the association between low preoperative body weight and postoperative fever was based on a single study [43]; this result may coincide and requires further validation.

Additionally, preoperative overweight status was associated with an increased risk of wound infections, intestinal fistulas, arrhythmias, and pancreatic fistulas, likely due to excess intra-abdominal fat. Lastly, age and sex might affect the association between preoperative BMI, post-operative complications, and survival outcomes, This could be explained as follows. (1) Older individuals are more prone to complications and face higher surgical risks compared to youn-ger patients. Additionally, younger patients typically respond better to treatment and have a greater capacity for physical recovery than older ones do; (2) male patients are more prone to develop GC than female patients, and their prognosis tends to be less favorable, primarily due to differences in hormone levels, biochemical mechanisms, and healthcare-seeking behaviors between men and women [63].

This study has certain limitations. First, all the included studies were retrospective in nature, which may have been influenced by recall and uncontrollable confounding biases. Sec-ond, the analysis of postoperative complications was based on crude data, and other factors that may affect postoperative complications were not adjusted for. This could have affected the risk of postoperative complications in relation to preoperative BMI. Third, the association between preoperative BMI and survival outcomes in patients was based on adjusted HR and 95%CIs; however, the factors adjusted for in the included studies were not consistent, which may have introduced an uncontrollable confounding bias. Fourth, we adopted the universal BMI classification criteria; however, the impact of overweight and underweight populations on the prognosis of cancer patients differs between Eastern and Western countries. This varia-tion influences the effect of the preoperative BMI on the prognosis of patients with GC. Fifth, tumor staging and postoperative chemotherapy regimens significantly influenced the progno-sis of patients with GC; however, detailed stratified data to further investigate the association between preoperative BMI and the prognosis of patients with GC were not available. Sixth, the association of preoperative BMI with the prognosis of patients with GC might be affected by fat mass, muscle mass, and waist circumference; however, this information was not available in the included studies. Seventh, weight-related metabolic or cardiovascular complications in overweight or obese patients with GC may affect surgical procedures, postoperative complica-tions, and survival outcomes. Nonetheless, the included studies did not provide stratified data on weight-related metabolic or cardiovascular complications. Finally, this study was based on a meta-analysis of published literature, and because the analysis was based on pooled data, there was inevitable publication bias and limitations in the exploratory analysis.

## Conclusions

The study identified a significant association between preoperative underweight status, higher risk of postoperative complications, and poorer survival outcomes. Additionally, preoperative overweight status was significantly associated with a higher risk of postoperative complications but improved survival outcomes in patients with GC. Therefore, a large-scale prospective study is needed to validate our findings.

## Supporting information

**S1 Checklist. PRISMA 2020 checklist.**
(DOCX)

**S1 Fig. Sensitivity for the association of preoperative underweight with the risk of postop-erative complications.**
(TIF)

**S2 Fig. Sensitivity for the association of preoperative overweight with the risk of postoperative complications.**
(TIF)

**S3 Fig. Funnel plot for the association of preoperative underweight with the risk of postoperative complications.**
(TIF)

**S4 Fig. Funnel plot for the association of preoperative overweight with the risk of postoperative complications.**
(TIF)

**S5 Fig. Sensitivity for the association of preoperative underweight with overall survival.**
(TIF)

**S6 Fig. Sensitivity for the association of preoperative overweight with overall survival.**
(TIF)

**S7 Fig. Funnel plot for the association of preoperative underweight with overall survival.**
(TIF)

**S8 Fig. Funnel plot for the association of preoperative overweight with overall survival.**
(TIF)

**S9 Fig. Sensitivity for the association of preoperative underweight with disease free survival.**
(TIF)

**S10 Fig. Sensitivity for the association of preoperative overweight with disease free survival.**
(TIF)

**S11 Fig. Funnel plot for the association of preoperative underweight with disease free survival.**
(TIF)

**S12 Fig. Funnel plot for the association of preoperative overweight with disease free survival.**
(TIF)

**S1 Table. Quality scores of prospective cohort studies using Newcastle-Ottawa Scale.**
(DOC)

**S1 Data.**
(XLSX)

**S2 Data.**
(XLSX)

## Author Contributions

**Conceptualization:** Zhenzhen Li, Lili Cui.

**Data curation:** Zhenzhen Li, Lili Cui.

**Formal analysis:** Zhenzhen Li, Lili Cui, Jing Sun, Wanlu Liu.

**Investigation:** Zhenzhen Li, Jing Sun, Wanlu Liu.

**Methodology:** Zhenzhen Li.

**Supervision:** Lili Cui, Jing Sun.

**Writing – original draft:** Zhenzhen Li, Lili Cui.

**Writing – review & editing:** Jing Sun, Wanlu Liu.

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
