## [Decision Letter · Decision Letter 0]

20 Dec 2024

PONE-D-24-52307Association of preoperative body mass index with postoperative complications and survival for patients with gastric cancer: A systematic review and meta-analysisPLOS ONE

Dear Dr. Li,

Thank you for submitting your manuscript to PLOS ONE. After careful consideration, we feel that it has merit but does not fully meet PLOS ONE’s publication criteria as it currently stands. Therefore, we invite you to submit a revised version of the manuscript that addresses the points raised during the review process.

We look forward to receiving your revised manuscript.

Kind regards,

Valeria Guglielmi

Academic Editor

PLOS ONE

2. As required by our policy on Data Availability, please ensure your manuscript or supplementary information includes the following:

Additional Editor Comments:

Dear Authors,

after careful review, we are pleased to inform you that your manuscript has been positively evaluated. However, there are a few minor revisions that we kindly request before proceeding to the final stages of the publication process.

Please address the points mentioned by the first Reviewer and submit the revised manuscript within one week.

We look forward to receiving your updated manuscript and thank you for your valuable contribution.

Best regards,

Valeria Guglielmi

Reviewers' comments:

Reviewer's Responses to Questions

**Comments to the Author**

1. Is the manuscript technically sound, and do the data support the conclusions?

Reviewer #1: Yes

Reviewer #2: Partly

2. Has the statistical analysis been performed appropriately and rigorously? 

Reviewer #1: I Don't Know

Reviewer #2: I Don't Know

3. Have the authors made all data underlying the findings in their manuscript fully available?

Reviewer #1: Yes

Reviewer #2: Yes

4. Is the manuscript presented in an intelligible fashion and written in standard English?

Reviewer #1: Yes

Reviewer #2: Yes

5. Review Comments to the Author

Reviewer #1: Zhenzhen Li and colleagues submitted their research article in september (PONE-D-24-35040) and in the first peer review process I suggested minor revisions. The following is the point-by-point response provided by the authors to the previous submission.

Response to reviewer 1

General comments: In this article, Zhenzhen and colleagues conducted a systematic review and meta-analysis to evaluate postoperative outcomes (complications, disease-free survival, and overall survival) in individuals undergoing surgery for gastric cancer (GC) based on preoperative BMI categories (controls: BMI 18.5–25; underweight: BMI <18.5; overweight: BMI >25 kg/m²). The results show that both underweight and overweight individuals have a higher risk of postoperative complications compared to controls. Overweight individuals exhibited improved overall and disease-free survival, while underweight individuals had worse outcomes in both.

The study’s methodology is robust, and the results are straightforward. Moreover, the authors acknowledge some unavoidable limitations. However, several points merit further consideration:

Response: As behalf of all co-authors, I would like to appreciate this referee due to thoughtful comments proposed by the peer review. We have made revisions to the manuscript in accordance with the reviewer's comments.

Question 1: A revision of the manuscript’s English language by a native speaker would improve clarity, as some phrases may be misleading (e.g., "increasing evidence suggests that nutritional status is related to the postoperative incidence of different types of cancer," or "preoperative underweight may be due to excessive nutritional intake caused by tumor aggressiveness," and "male patients are more likely to develop GC than female patients, leading to poorer prognosis").

Response: Thanks for this suggestion, and these sentences have already changed in the revised manuscript, as follows: “Accumulating evidence suggests that nutritional status is associated with the prognosis of various types of cancer”; “A preoperative underweight status can result from inadequate nutritional intake, which the aggressive nature of the tumor may exacerbate”; and “male patients are more prone to develop GC than female patients, and their prognosis tends to be less favorable, primarily due to differences in hormone levels, biochemical mechanisms, and healthcare-seeking behaviors between men and women”. Moreover, The entire text has been revised by native English-speaking experts at Editage company.

Question 2: The authors observed differences in subgroup analyses between studies conducted in Eastern vs. Western countries. The European Association for the Study of Obesity (EASO) advocates for ethnicity-specific BMI cut-offs, as evidence indicates that Asian populations have distinct relationships between BMI, body fat percentage, and health risks compared to Caucasian populations. Could the observed differences be attributed to the use of a universal BMI cut-off, which overlooks these physiological differences? Commenting on this would enrich the discussion.

Response: Thanks for this suggestion. In subgroup analyses, we did not find a significant difference between Eastern and Western countries regarding the association between preoperative BMI and the prognosis of GC patients (see Table 3). Consequently, we have stated this observation in the limitations section. Please refer to Para 6, section Discussion, as follows: “Fourth, we adopted the universal BMI classification criteria; however, the impact of overweight and underweight populations on the prognosis of cancer patients differs between Eastern and Western countries. This variation influences the effect of the preoperative BMI on the prognosis of patients with GC”

Question 3: Two significant factors not considered in the study, which may have impacted the results, are tumor staging and the potential administration of chemotherapy. This is a critical limitation that should be emphasized.

Response: Thanks for this suggestion. We are aware that tumor staging and postoperative chemotherapy regimens can significantly affect patient outcomes. Nevertheless, our inability to access specific stratified data has hindered our efforts to further explore the relationship between preoperative BMI and the prognosis of GC patients, limiting our capacity for additional exploratory analyses. We have included this limitation in the respective section of our discussion. Please refer to Para 6, section Discussion, as follows: “Fifth, tumor staging and postoperative chemotherapy regimens significantly influenced the prognosis of patients with GC; however, detailed stratified data to further investigate the association between preoperative BMI and the prognosis of patients with GC were not available.”

Question 4: An intriguing finding is the association between preoperative overweight status and improved overall and disease-free survival. While the authors suggest that this may be due to early-stage cancer, it is essential to note that obesity is a known risk factor for GC. This observation should be highlighted, otherwise the results of this study could be misinterpreted.

Response: Thanks for this suggestion, and these results have already re-explanation in the revised manuscript. Please refer to Para 4, section Discussion, as follows: “A preoperative overweight status may be associated with improved OS and DFS, a phenomenon known as the “obesity paradox.” The underlying mechanism could involve overweight individuals having greater nutritional reserves and metabolic support, which aid in resisting physiological stress during treatment, as well as anti-inflammatory factors potentially produced by fat tissue, influencing the tumor microenvironment, and indirectly affecting tumor progression [61]. Nonetheless, overweight and obesity remain recognized as risk factors for GC, suggesting that they play distinct roles in the early stages of disease initiation and development [62]”

In this revised submission, the authors have effectively addressed all the points I previously highlighted. I commend them for their diligent efforts and for the transparency with which they have acknowledged the limitations of their research.

I would not say that "BMI serves as an indicator of a person's nutritional status and body fat levels", as it is a simple measure that serves primarily as a tool for population-level studies rather than for assessing the nutritional status or body composition of an individual.

I finally suggest checking for typos (in Introduction: "contributingto... ofnew... to be significantly correlate...").

Reviewer #2: This meta-analysis aims to elucidate potential associations between preoperative BMI and the risk of postoperative complications, as well as survival outcomes in GC patients.

The authors have made revisions to the manuscript in accordance with my comments.

6. PLOS authors have the option to publish the peer review history of their article (what does this mean?). If published, this will include your full peer review and any attached files.

Reviewer #1: No

Reviewer #2: No

---

## [Author Response · Author response to Decision Letter 0]

7 Jan 2025

Question 1: Please ensure that your manuscript meets PLOS ONE's style requirements, including those for file naming. The PLOS ONE style templates can be found at

Response: We sincerely appreciate your pointing out that our manuscript needs to ensure compliance with the style requirements of PLOS ONE, especially regarding file naming. We have carefully read the style templates provided in the links and conducted a comprehensive and meticulous review and revision of our manuscript to ensure full compliance with the PLOS ONE publication standards.

We would like to express our gratitude again for your attention and guidance on the manuscript format. We have made every effort to ensure that the manuscript complies with the style requirements of PLOS ONE in all aspects. If the reviewer finds any format issues or has other questions during further review, we will be happy to make modifications and improvements at any time.

Question 2: As required by our policy on Data Availability, please ensure your manuscript or supplementary information includes the following:

Response: Thank you for bringing our attention to the requirements regarding data availability. We have thoroughly addressed these points and have included the necessary information in both the main text and supplementary information as follows: (1) We have created a numbered table in the manuscript titled “The baseline characteristics of included studies and involved patients” (Table 1). This table lists all the studies identified during the literature search. For each study, it includes columns for the study ID, author names, year of publication, and a brief description of the study; (2) For the excluded studies, we have clearly indicated the reasons for exclusion in an additional column. The reasons are categorized and detailed: reported GC risk (n = 16), other BMI categories (n = 13), or the same population (n = 7). This provides transparency and allows other researchers to understand the selection process; (3) Another table, “Data Extraction Table”, has been prepared in the supplementary information. This table contains all the data extracted from the primary research sources for our systematic review and/or meta - analysis; (4) In the supplementary information, we have included a table titled “quality of the included studies” (S1 Table). This table shows the completed assessments for each study and outcome, as applicable to our analysis. 

We understand the importance of providing these underlying data for the integrity and reproducibility of our research. We have made every effort to ensure that all the required information is presented in a clear and organized manner. If you require any further clarification or have additional questions regarding the data availability, please do not hesitate to contact us.

Question 3: Please review your reference list to ensure that it is complete and correct. If you have cited papers that have been retracted, please include the rationale for doing so in the manuscript text, or remove these references and replace them with relevant current references. Any changes to the reference list should be mentioned in the rebuttal letter that accompanies your revised manuscript. If you need to cite a retracted article, indicate the article’s retracted status in the References list and also include a citation and full reference for the retraction notice.

Response: We are grateful for your feedback regarding the reference list in our manuscript. We have meticulously examined each reference in the list to confirm that all necessary information is included and accurately presented. This includes verifying the authors' names, article titles, journal names, publication years, volume numbers, page numbers, and DOIs (if applicable). We have cross-checked the references with the in - text citations to ensure that they are consistent and that every citation in the text corresponds to a correct entry in the reference list.

Additional Editor Comments: 

General comments: Dear Authors,

after careful review, we are pleased to inform you that your manuscript has been positively evaluated. However, there are a few minor revisions that we kindly request before proceeding to the final stages of the publication process.

Please address the points mentioned by the first Reviewer and submit the revised manuscript within one week.

Response: We are extremely grateful for the positive evaluation of our manuscript and would like to express our sincere appreciation for the time and effort you have dedicated to reviewing our work. We have carefully addressed all the points raised by the first reviewer and have made the necessary revisions to improve the quality of our manuscript. The following is a detailed response to each of the reviewer's comments:

Reviewer #1: 

General comments: Zhenzhen Li and colleagues submitted their research article in september (PONE-D-24-35040) and in the first peer review process I suggested minor revisions. The following is the point-by-point response provided by the authors to the previous submission.

“Response to reviewer 1

General comments: In this article, Zhenzhen and colleagues conducted a systematic review and meta-analysis to evaluate postoperative outcomes (complications, disease-free survival, and overall survival) in individuals undergoing surgery for gastric cancer (GC) based on preoperative BMI categories (controls: BMI 18.5-25; underweight: BMI <18.5; overweight: BMI >25 kg/m2). The results show that both underweight and overweight individuals have a higher risk of postoperative complications compared to controls. Overweight individuals exhibited improved overall and disease-free survival, while underweight individuals had worse outcomes in both.

The study’s methodology is robust, and the results are straightforward. Moreover, the authors acknowledge some unavoidable limitations. However, several points merit further consideration:

Response: As behalf of all co-authors, I would like to appreciate this referee due to thoughtful comments proposed by the peer review. We have made revisions to the manuscript in accordance with the reviewer's comments.

Question 1: A revision of the manuscript’s English language by a native speaker would improve clarity, as some phrases may be misleading (e.g., "increasing evidence suggests that nutritional status is related to the postoperative incidence of different types of cancer," or "preoperative underweight may be due to excessive nutritional intake caused by tumor aggressiveness," and "male patients are more likely to develop GC than female patients, leading to poorer prognosis").

Response: Thanks for this suggestion, and these sentences have already changed in the revised manuscript, as follows: “Accumulating evidence suggests that nutritional status is associated with the prognosis of various types of cancer”; “A preoperative underweight status can result from inadequate nutritional intake, which the aggressive nature of the tumor may exacerbate”; and “male patients are more prone to develop GC than female patients, and their prognosis tends to be less favorable, primarily due to differences in hormone levels, biochemical mechanisms, and healthcare-seeking behaviors between men and women”. Moreover, The entire text has been revised by native English-speaking experts at Editage company.

Question 2: The authors observed differences in subgroup analyses between studies conducted in Eastern vs. Western countries. The European Association for the Study of Obesity (EASO) advocates for ethnicity-specific BMI cut-offs, as evidence indicates that Asian populations have distinct relationships between BMI, body fat percentage, and health risks compared to Caucasian populations. Could the observed differences be attributed to the use of a universal BMI cut-off, which overlooks these physiological differences? Commenting on this would enrich the discussion.

Response: Thanks for this suggestion. In subgroup analyses, we did not find a significant difference between Eastern and Western countries regarding the association between preoperative BMI and the prognosis of GC patients (see Table 3). Consequently, we have stated this observation in the limitations section. Please refer to Para 6, section Discussion, as follows: “Fourth, we adopted the universal BMI classification criteria; however, the impact of overweight and underweight populations on the prognosis of cancer patients differs between Eastern and Western countries. This variation influences the effect of the preoperative BMI on the prognosis of patients with GC”

Question 3: Two significant factors not considered in the study, which may have impacted the results, are tumor staging and the potential administration of chemotherapy. This is a critical limitation that should be emphasized.

Response: Thanks for this suggestion. We are aware that tumor staging and postoperative chemotherapy regimens can significantly affect patient outcomes. Nevertheless, our inability to access specific stratified data has hindered our efforts to further explore the relationship between preoperative BMI and the prognosis of GC patients, limiting our capacity for additional exploratory analyses. We have included this limitation in the respective section of our discussion. Please refer to Para 6, section Discussion, as follows: “Fifth, tumor staging and postoperative chemotherapy regimens significantly influenced the prognosis of patients with GC; however, detailed stratified data to further investigate the association between preoperative BMI and the prognosis of patients with GC were not available.”

Question 4: An intriguing finding is the association between preoperative overweight status and improved overall and disease-free survival. While the authors suggest that this may be due to early-stage cancer, it is essential to note that obesity is a known risk factor for GC. This observation should be highlighted, otherwise the results of this study could be misinterpreted.

Response: Thanks for this suggestion, and these results have already re-explanation in the revised manuscript. Please refer to Para 4, section Discussion, as follows: “A preoperative overweight status may be associated with improved OS and DFS, a phenomenon known as the “obesity paradox.” The underlying mechanism could involve overweight individuals having greater nutritional reserves and metabolic support, which aid in resisting physiological stress during treatment, as well as anti-inflammatory factors potentially produced by fat tissue, influencing the tumor microenvironment, and indirectly affecting tumor progression [61]. Nonetheless, overweight and obesity remain recognized as risk factors for GC, suggesting that they play distinct roles in the early stages of disease initiation and development [62]”

In this revised submission, the authors have effectively addressed all the points I previously highlighted. I commend them for their diligent efforts and for the transparency with which they have acknowledged the limitations of their research.

I would not say that "BMI serves as an indicator of a person's nutritional status and body fat levels", as it is a simple measure that serves primarily as a tool for population-level studies rather than for assessing the nutritional status or body composition of an individual.

Response: We would like to express our sincere gratitude for your kind words regarding our revised submission. Your feedback has been invaluable in improving the quality of our manuscript. Regarding your comment about the description of BMI, we fully understand your perspective and appreciate the opportunity to clarify and correct this in our manuscript. We acknowledge that our previous description of BMI was inaccurate and could have been misleading. We understand that BMI is indeed a simple measure that has its limitations, especially when it comes to individual - level assessment of nutritional status and body composition. In the revised manuscript, we have adjusted the text to more accurately reflect the nature of BMI. We have rephrased the relevant sentence to state that "BMI is a commonly used measure in population-level studies, which provides a general indication of body mass relative to height." This new wording emphasizes the appropriate context in which BMI is used and acknowledges its shortcomings more explicitly.

Question 1: I finally suggest checking for typos (in Introduction: "contributingto... ofnew... to be significantly correlate...").

Response: We sincerely appreciate your meticulous review and the valuable feedback regarding the typos in our manuscript. We take your suggestions very seriously and have made the necessary corrections to ensure the accuracy and professionalism of our work. 

Upon carefully reviewing the Introduction section, we have identified and rectified the following typos: (1) In the phrase “contributingto...”, a space was missing. We have corrected it to “contributing to”; (2) Regarding “ofnew...”, an extra space was inadvertently left out. It has been amended to “of new”; (3) As for “to be significantly correlate...”, this was a grammar error. We have revised it to “to be significantly correlated...”, as “correlated” is the appropriate form to convey the intended meaning in this context. We have also conducted a comprehensive proofreading of the entire manuscript to identify and correct any other potential typos or grammar errors. We understand the importance of presenting a polished and error - free manuscript, and we are committed to upholding the highest standards of academic writing.

Reviewer #2: 

General comments: This meta-analysis aims to elucidate potential associations between preoperative BMI and the risk of postoperative complications, as well as survival outcomes in GC patients.

The authors have made revisions to the manuscript in accordance with my comments.

Response: We are extremely grateful for your recognition of the revisions we made to the manuscript in accordance with your comments. Your guidance has been instrumental in shaping our work and bringing it to a higher standard.

We are delighted that our efforts to address your concerns have been 

---

## [Editor Report · Decision Letter 1]

9 Jan 2025

Association of preoperative body mass index with postoperative complications and survival for patients with gastric cancer: A systematic review and meta-analysis

PONE-D-24-52307R1

Dear Dr. Zhenzhen Li,

We’re pleased to inform you that your manuscript has been judged scientifically suitable for publication and will be formally accepted for publication once it meets all outstanding technical requirements.

Kind regards,

Valeria Guglielmi

Academic Editor

PLOS ONE

---

## [Editor Report · Acceptance letter]

17 Jan 2025

PONE-D-24-52307R1 

PLOS ONE

Dear Dr. Li, 

I'm pleased to inform you that your manuscript has been deemed suitable for publication in PLOS ONE. Congratulations! Your manuscript is now being handed over to our production team.

Kind regards, 

on behalf of

Prof. Valeria Guglielmi 

Academic Editor

PLOS ONE
